# SARS-CoV-2 Omicron Variant Neutralization after Third Dose Vaccination in PLWH

**DOI:** 10.3390/v14081710

**Published:** 2022-08-03

**Authors:** Alessandra Vergori, Alessandro Cozzi-Lepri, Giulia Matusali, Francesca Colavita, Stefania Cicalini, Paola Gallì, Anna Rosa Garbuglia, Marisa Fusto, Vincenzo Puro, Fabrizio Maggi, Enrico Girardi, Francesco Vaia, Andrea Antinori

**Affiliations:** 1HIV/AIDS Unit, National Institute for Infectious Diseases Lazzaro Spallanzani IRCCS, 00149 Rome, Italy; stefania.cicalini@inmi.it (S.C.); marisa.fusto@inmi.it (M.F.); andrea.antinori@inmi.it (A.A.); 2Centre for Clinical Research, Epidemiology, Modelling and Evaluation (CREME), Institute for Global Health, UCL, London WC1E 6BT, UK; a.cozzi-lepri@ucl.ac.uk; 3Laboratory of Virology, National Institute for Infectious Diseases Lazzaro Spallanzani IRCCS, 00149 Rome, Italy; giulia.matusali@inmi.it (G.M.); francesca.colavita@inmi.it (F.C.); annarosa.garbuglia@inmi.it (A.R.G.); fabrizio.maggi@inmi.it (F.M.); 4Health Direction, National Institute for Infectious Diseases Lazzaro Spallanzani IRCCS, 00149 Rome, Italy; paola.galli@inmi.it; 5Risk Management Unit, National Institute for Infectious Diseases Lazzaro Spallanzani IRCCS, 00149 Rome, Italy; vincenzo.puro@inmi.it; 6Scientific Direction, National Institute for Infectious Diseases Lazzaro Spallanzani IRCCS, 00149 Rome, Italy; enrico.girardi@inmi.it; 7General Direction, National Institute for Infectious Diseases Lazzaro Spallanzani IRCCS, 00149 Rome, Italy; francesco.vaia@inmi.it

**Keywords:** SARS-CoV-2, HIV/AIDS, SARS-CoV-2 vaccine, Omicron variant, neutralization titers, third dose vaccine

## Abstract

The aim was to measure neutralizing antibody levels against the SARS-CoV-2 Omicron (BA.1) variant in serum samples obtained from vaccinated PLWH and healthcare workers (HCW) and compare them with those against the Wuhan-D614G (W-D614G) strain, before and after the third dose of a mRNA vaccine. We included 106 PLWH and 28 HCWs, for a total of 134 participants. Before the third dose, the proportion of participants with undetectable nAbsT against BA.1 was 88% in the PLWH low CD4 nadir group, 80% in the high nadir group and 100% in the HCW. Before the third dose, the proportion of participants with detectable nAbsT against BA.1 was 12% in the PLWH low nadir group, 20% in the high nadir group and 0% in HCW, respectively. After 2 weeks from the third dose, 89% of the PLWH in the low nadir group, 100% in the high nadir group and 96% of HCW elicited detectable nAbsT against BA.1. After the third dose, the mean log2 nAbsT against BA.1 in the HCW and PLWH with a high nadir group was lower than that seen against W-D614G (6.1 log2 (±1.8) vs. 7.9 (±1.1) and 6.4 (±1.3) vs. 8.6 (±0.8)), respectively. We found no evidence of a different level of nAbsT neutralization by BA.1 vs. W-D614G between PLWH with a high CD4 nadir and HCW (0.40 (−1.64, 2.43); *p* = 0.703). Interestingly, in PLWH with a low CD4 nadir, the mean log2 difference between nAbsT against BA.1 and W-D614G was smaller in those with current CD4 counts 201–500 vs. those with CD4 counts < 200 cells/mm^3^ (−0.80 (−1.52, −0.08); *p* = 0.029), suggesting that in this target population with a low CD4 nadir, current CD4 count might play a role in diversifying the level of SARS-CoV-2 neutralization.

## 1. Introduction

Persons living with HIV (PLWH) might have an increased risk of adverse outcomes following COVID-19 infection [1]. The high contagiousness and spread of the Omicron (BA.1 and BA.2) variant of SARS-CoV-2 and its ability to evade immunity elicited by vaccination or infection are of increasing concern. We previously showed that, in PLWH, after three doses of the COVID-19 mRNA vaccine, the humoral response elicited was strong and higher than that achieved with the second dose (>2 log_2_ difference), and neutralizing antibodies (nAbs) against the Whuan-D614G strain SARS-CoV-2 increased in most of the participants regardless of their CD4 count at the time of first dose vaccination [2]. In this work, we aimed to evaluate the potential susceptibility of the Omicron BA.1 variant to the mRNA vaccine and to compare neutralization titers in PLWH with those observed in a control sample of health care workers.

## 2. Materials and Methods

In a subset of vaccinated PLWH participating in the HIV-VAC study (details described elsewhere) [3], we measured levels of neutralization to BA.1 and Wuhan-D614G (W-D614G) in stored serum samples collected before and after the third dose. An external control group of HIV-negative health care workers (HCWs) vaccinated with the third booster dose (BD) was also included for comparison. The study was approved by the Scientific Committee of the Italian Drug Agency (AIFA) and by the Ethical Committee of the Lazzaro Spallanzani Institute, as the National Review Board for COVID-19 pandemic in Italy (approval number 323/2021). PLWH were firstly stratified according to their CD4 count nadir (<350 (low nadir group) vs. >350 cells/mm^3^ (high nadir group)). In addition, the group with low nadir CD4 count (<350/mm^3^) was further stratified according to the CD4 count at the time of the booster vaccine dose (>200/mm^3^, 201–500/mm^3^ and >500 cells/mm^3^), which had increased as a result of treatment with ART. All participants received either an additional 3rd dose (full dose at least 28 days after the 2nd, PLWH, low nadir group) or a booster dose of vaccine (booster at least 5 months after the 2nd, high nadir and HCW groups). Neutralizing antibody titers (nAbsT) were measured by micro-neutralization assay based on live SARS-CoV-2 virus (described elsewhere [4]) for W-D614G (Ref-SKU: 008V–04005, from EVAg portal), and BA.1 (GISAID accession ID EPI_ISL_7716384), before and after the 3rd dose (after 15 days in PLWH and 30 days in HCW). The highest serum dilution inhibiting at least 90% of the cytopathic effect on Vero E6 cells was defined as neutralizing, and nAbs were categorized as undetectable if titers were <1:10. Proportions with detectable responses were compared using the McNemar’s test for paired data and chi-square and Fisher exact test for unpaired data. Mean levels of nAbsT to BA.1 vs. W-D614G (in the log2 scale) were compared within groups using a paired *t*-test and across groups using a truncated linear regression model (to account for censored response data) after controlling for gender, age and time elapsed since the end of the primary vaccination cycle.

## 3. Results

We included 106 PLWH, of whom there were 81 in the low CD4 nadir group (27 (33%) with a CD4 count < 200/mm^3^, 29 (36%) with a CD4 count 201–500/mm^3^ and 25 (31%) with a CD4 count > 500/mm^3^), and 25 in the high nadir group and 28 HCWs, for a total of 134 participants (characteristics shown in Appendix A).

At the time of receiving the third dose, the proportion of participants with detectable nAbsT against Whuan-D614G and BA.1 was 59 % and 12 % in the PLWH low nadir group (*p* < 0.0001) after a median of 156 days (IQR 152–159, min 4.4 months) after completion of the primary cycle (after the two-dose vaccine cycle), 80% and 20% in the high nadir group (after a median of 182 (177–186) days, min 4.9 months) (*p* = 0.0001) and 64% and 0% in the HCW (after a median of 283 (278–28) days, min 8.9 months), respectively.

After 2 weeks from receiving the third dose, 89% of PLWH in the low nadir group, 100% in the high nadir group and 96% of HCW elicited detectable nAbsT against BA.1 (*p* = 0.123). At the same time, the proportion of the participants with detectable nABsT against Whuan-D614G was 94%, 100% and 100%, respectively (*p* = 0.182).

After the third dose, the mean log2 nAbsT against BA.1 in the HCW and PLWH with a high nadir was lower than that seen against W-D614G (6.1 log_2_ (±1.8) vs. 7.9 (±1.1) and 6.4 (±1.3) vs. 8.6 (±0.8), respectively) (*p =* < 0.0001, Figure 1A). However, from fitting a truncated linear regression, we found no evidence of a different level of nAbsT neutralization by BA.1 vs. W-D614G between PLWH with a high CD4 nadir and HCW (0.40 (−1.64, 2.43); *p =* 0.70, Appendix A, panel A).

As well, the mean log_2_ nAbsT against BA.1 (vs. D614G) in PLWH with a high nadir and those with low nadir was 6.4 log_2_ (±1.3) vs. 8.6 (±0.8) and 6.3 (±2.1) vs. 8.4 (±2.4), respectively, (Figure 1B). From fitting a truncated linear regression, we found no evidence of a different level of nAbsT neutralization by BA.1 vs. W-D614G between PLWH with high and low CD4 nadir (0.23 (−0.44, 0.91); *p =* 0.497, Appendix A, panel B).

In contrast, even after controlling for potential confounding bias and censored responses by regression adjustment, among PLWH with a low CD4 nadir, the mean log_2_ difference between nAbsT against BA.1 and WD614G was smaller in those with a current CD4 count 201–500 vs. those with a CD4 count < 200 cells/mm^3^ (−0.80 (−1.52, −0.08); *p =* 0.03) (Figure 1B, Appendix A, panel C).

## 4. Discussion

Our data show that neutralizing activity against BA.1 strongly increased after a third dose of a mRNA vaccine in all participants but was poorer than that seen against the original W-D614G strain, regardless of HIV status. This is likely due to the fact that the original mRNA vaccines were developed against the original strain, so they are less protective against current circulating variants [4]. There are limitations to this analysis. First, we only evaluated the short-term response (2 weeks on average) to the third dose and, therefore, we are unable to provide estimates of the durability and waning of nAbsT. Second, our analysis was not powered to establish whether the nAbsT recovering after a third dose was also associated with a reduced incidence of infection or severe disease. Third, nAbs activity against the past (e.g., Delta) and most recent variants (e.g., BA.1.1, BA.2) was not measured, although it has been recently demonstrated that the immune escape exhibited by all Omicron sublineages (BA.1.1 and BA.2) seems largely overcome by the booster dose compared with the two-dose cycle, suggesting a certain degree of cross-reactive immunity [5]. Last but not least, our results are only valid under the usual strong assumptions of a correctly specified model (liner regression predictor), the inclusion of all key potential confounding factors (age, gender and time since completion of primary cycle) and no unmeasured confounding factors being present. In particular, the fact that vaccination kinetics in the low CD4 count group vs. HCW was largely different, as per AIFA recommendations (although down to a maximum of one month in our sample) might have introduced residual confounding bias. Moreover, reasons for not seeing a clear dose-response relationship with CD4 count level (as there was no evidence of a difference between participants with a CD4 count < 200 and those with a CD4 count > 500 and the HCW) are unclear.

## 5. Conclusions

In conclusion, our results show that after at least 5 months from a primary two-dose vaccination cycle and shortly after the third dose, both with one additional and a booster, neutralizing activity against BA.1 strongly increased in all participants but was poorer than that seen against the original W-D614G strain, regardless of HIV status. Although, we found a signal for an association between CD4 count level and the extent of neutralization, and this should be further evaluated in future studies. These results are useful to appropriately define future boosting vaccination strategies in PLWH, accounting for the currently circulating SARS-CoV-2 variant.

## Figures and Tables

**Figure 1 viruses-14-01710-f001:**
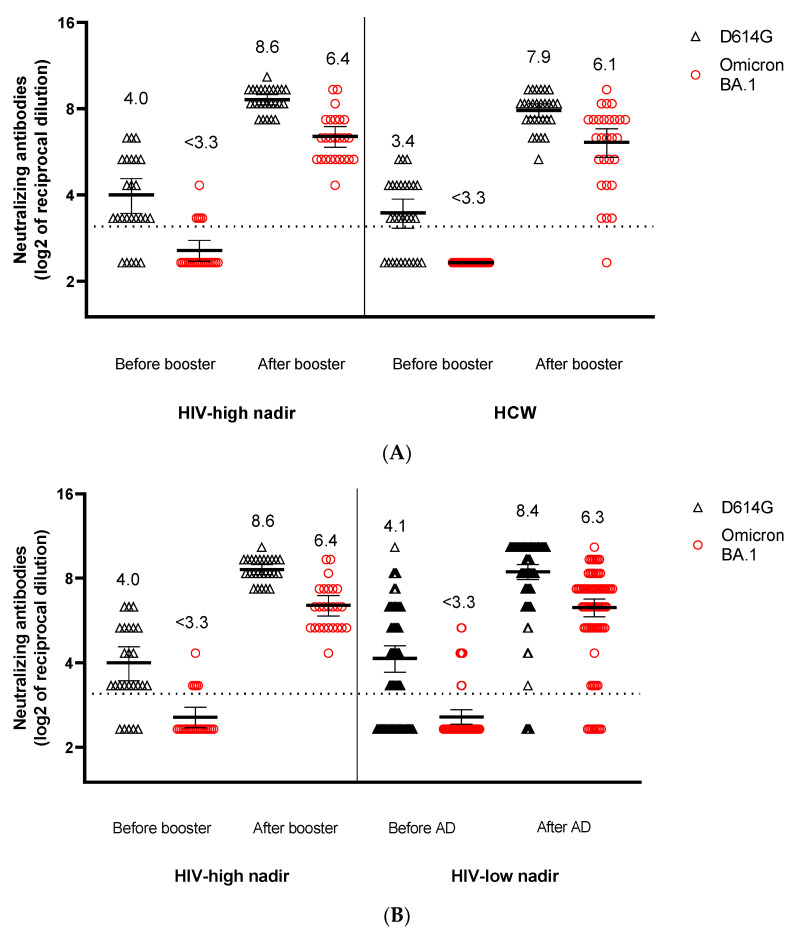
Log_2_ mean titers of neutralizing antibodies against Wuhan-D614G and BA.1 in (**A**) PLWH with high CD4 count/high nadir and health care workers (HIV-negative); (**B**) PLWH with low and high CD4 nadir; (**C**) PLWH with low CD4 nadir and with CD4 count < 200/mm^3^, 201–500/mm^3^, >501/mm^3^ at the time of third vaccine dose.

## Data Availability

Anonymized participant data will be made available upon reasonable requests directed to the corresponding author. Proposals will be reviewed and approved by the investigator and collaborators on the basis of scientific merit. After approval of a proposal, data can be shared through a secure online platform after signing a data access agreement.

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
