# Peer review of "SARS-CoV-2 Omicron Variant Neutralization after Third Dose Vaccination in PLWH"

_viruses, 2022, doi:10.3390/v14081710_

Round 1
Reviewer 1 Report
Because persons living with HIV (PLWH) might have an increased risk of adverse outcomes following COVID-19, it is important to research neutralizing antibodies levels against SARS-CoV-2 in serum samples obtained from vaccinated PLWH and HCW.
Comment 1:
There are inconsistencies in the following statements.
“PLWH were stratified according to their CD4 count nadir [<350 (low nadir group) vs. >350 cells/mm3 (high nadir group)]. The low nadir group was further stratified according to the current CD4 count (>200/mm3, CD4 201-500/mm3 and >500 cells/mm3)”
Comment 2:
The author said that the neutralizing activity against W-D614G was stronger than the neutralizing activity against BA.1. Could you mention the neutralizing activity against any other variant in the discussion part?
Comment 3:
Throughout, it is difficult to know whether you are comparing PLWH and HCW or D614G and BA1. In other words What is the most important point in this paper? Please consider improving this if possible.
Author Response
Reviewer #1
Because persons living with HIV (PLWH) might have an increased risk of adverse outcomes following COVID-19, it is important to research neutralizing antibodies levels against SARS-CoV-2 in serum samples obtained from vaccinated PLWH and HCW.
Comment 1:
There are inconsistencies in the following statements.
“PLWH were stratified according to their CD4 count nadir [<350 (low nadir group) vs. >350 cells/mm3 (high nadir group)]. The low nadir group was further stratified according to the current CD4 count (>200/mm3, CD4 201-500/mm3 and >500 cells/mm3)”
The current value of CD4 count refer to a value measured temporally an average of 5 (IQR:2, 8) years after the data of the nadir value (time since the AIDS diagnosis) and after the participants had been receiving ART for some time. Therefore, for the average participant, the current CD4 count would have been greater than his/her CD4 count nadir value. To improve clarity, we have slightly rephrased this sentence in the revised method section as follows:
PLWH were firstly stratified according to their CD4 count nadir [<350 (low nadir group) vs. >350 cells/mm3(high nadir group)]. In addition, the group with low nadir CD4 count (<350/mm3) was further stratified according to the CD4 count at the time of the booster vaccine dose (>200/mm3, CD4 201-500/mm3 and >500 cells/mm3) which had increased as a result of treatment with ART.
Comment 2:
The author said that the neutralizing activity against W-D614G was stronger than the neutralizing activity against BA.1. Could you mention the neutralizing activity against any other variant in the discussion part?
We thank the reviewer for the interesting observation but unfortunately no other tests have been performed on these same samples at this stage. The neutralization tests were carried out during an epidemiological era in which the Omicron BA.1 variant prevailed. We therefore thought that it would have been most appropriate to test the neutralization against BA.1 and use W-D614G as a reference strain.
Comment 3:
Throughout, it is difficult to know whether you are comparing PLWH and HCW or D614G and BA1. In other words What is the most important point in this paper? Please consider improving this if possible.
In our mind both questions are highly relevant. We have now added a sentence in the Introduction to clarify the analysis objectives and that we did not have a specific ranking in mind with regards to these questions.
Reviewer 2 Report
In this article Vergori et al. analyze the induction of neutralizing antibodies before and after a third booster dose against Wuhan-D614G strain and omicron BA.1 in different groups of HIV-infected patients. In previous communication (CROI 2022 poster) the authors described an increase of neutralizing antibodies in all participants regardless of CD4 counts. This is an important topic because the potential differences in humoral responses among different groups of PLWH. In particular those patients that start treatment late after infection or with low CD4 nadir in which extensive damage of CD4 subsets as Tfh have been described the capacity to raise an appropriate immune response could be impaired.
However the article requires a major revision because writing is confusing and some conclusions are not properly justified.
Major comments
1. In the materials and methods section (lanes 55-58) it is stated that in the design of the study “All participants received either an additional 3rd dose (full dose at least 28 days after the 2nd, PLWH, low nadir group) or a booster dose of vaccine (booster at least 5 months after the 2nd , high nadir and HCW groups)” (lanes 55-58). Which is the justification of giving an additional 3rd dose with different kinetics?: 28 days after second dose in patients with low CD4 counts and 5 months later in patients with high CD4 counts? This is an important point that can bias the results because memory immune response is highly dependent on vaccine schedule and waning of antibodies depends on time after immunization. If time elapsed from 2nd to 3rd dose is very different among groups comparisons become difficult.
2. This description is contradictory with the results described in paragraph (lanes 73-76) in which it is said that patients received the 3rd dose at a median of 156 days “after completion of primary cycle” and with data provided in the supplementary table 2. Please clarify this point.
3. In the same paragraph it is described that in all groups the proportion of detectable neutralizing antibodies against BA.1 were much higher than against Wuhan-D614G which is nonsense. In the summary it is stated the contrary (lanes 19-20) as well as in the figure 1. Please clarify if this is a typing mistake and it should be “undetectable” instead of “detectable”. If it is the case, consider that this kind of mistakes made the task of the reviewer arduous to understand what the authors are saying.
4. In panel A of figure 1 it should be included the results of all patients included in the low nadir group for comparison. Panel B should remain to see the distribution among the three groups of patients with low CD4 nadir.
5. The sentence in lanes 87-90 “In contrast, in PLWH with low CD4 nadir the mean log2 difference between nAbsT against BA.1 and WD614G was smaller in those with current CD4 201-500 vs. those with CD4 <200 cells/mm3 [-0.80 (-1.52, -0.08); p=0.03] (Figure 1B, supplementary Table 2)” is difficult to understand. Difference between Wuhan and BA.1 neutralizing values seems not so different and follow the same trend. Please clarify this analysis and its relevance.
6. To conclude from previous analysis that “In contrast, in PLWH with a CD4 nadir <200 cells/mm3, current CD4 count appeared to play a role in diversifying the level of SARS CoV-2 neutralization” is not justified.
7. Differences in neutralization activity against original Wuhan and ómicron variants have been described in many works because the vaccine used is based on the sequence of the original SARS-CoV-2 strain and elicited antibodies recognize with less efficacy BA.1 that can be considered a escape variant. This should be discussed to explain such differences.
Minor comments
1. In the abstract please revise the last sentence for better understanding.
2. There is a mistake in materials and methods section (lane 12): >200 should be <200
Author Response
Reviewer #2
In this article Vergori et al. analyze the induction of neutralizing antibodies before and after a third booster dose against Wuhan-D614G strain and omicron BA.1 in different groups of HIV-infected patients. In previous communication (CROI 2022 poster) the authors described an increase of neutralizing antibodies in all participants regardless of CD4 counts. This is an important topic because the potential differences in humoral responses among different groups of PLWH. In particular those patients that start treatment late after infection or with low CD4 nadir in which extensive damage of CD4 subsets as Tfh have been described the capacity to raise an appropriate immune response could be impaired.
However the article requires a major revision because writing is confusing and some conclusions are not properly justified.
Major comments
- In the materials and methods section (lanes 55-58) it is stated that in the design of the study “All participants received either an additional 3rd dose (full dose at least 28 days after the 2nd, PLWH, low nadir group) or a booster dose of vaccine (booster at least 5 months after the 2nd , high nadir and HCW groups)” (lanes 55-58). Which is the justification of giving an additional 3rd dose with different kinetics? 28 days after second dose in patients with low CD4 counts and 5 months later in patients with high CD4 counts?
The difference in kinetics between PLWH and HCV is an inevitable source of bias. This is because Italian guidelines are different with regards to the provision of a 3rd dose in people with HIV and the general population. These are real world data reflecting Italian policy of vaccination in the two groups. Of note, the low nadir group were prioritized for vaccination and as a result a much longer time than 28 days had elapsed since the initiation of primary vaccination which played in our favor because the kinetics were a lot more balanced than expected.
- This is an important point that can bias the results because memory immune response is highly dependent on vaccine schedule and waning of antibodies depends on time after immunization. If time elapsed from 2nd to 3rd dose is very different among groups comparisons become difficult.
We agree with the reviewer that this is an important source of bias. We have tried to minimize the impact of this bias by including the time elapsed from 2nd dose to 3rd dose as a covariate in the truncated linear regression analysis. We have added a sentence in the revised Discussion section that despite our efforts residual confounding might be present due to the difference in vaccination kinetics.
- This description is contradictory with the results described in paragraph (lines 73-76) in which it is said that patients received the 3rd dose at a median of 156 days “after completion of primary cycle” and with data provided in the supplementary table 2. Please clarify this point.
We agree that the sentence was confusing and did not include the relevant data to check consistency. We have now more clearly reported in the revised text the median time from the end of the primary cycle separately by group and we also added the minimum number of months by group.
- In the same paragraph it is described that in all groups the proportion of detectable neutralizing antibodies against BA.1 were much higher than against Wuhan-D614G which is nonsense. In the summary it is stated the contrary (lines 19-20) as well as in the Figure 1. Please clarify if this is a typing mistake and it should be “undetectable” instead of “detectable”. If it is the case, consider that this kind of mistakes made the task of the reviewer arduous to understand what the authors are saying.
We apologies for the confusion. Indeed, the proportion of participants with detectable neutralizing antibodies were not correct and in the reverse order. This has now been corrected.
- In panel A of figure 1 it should be included the results of all patients included in the low nadir group for comparison. Panel B should remain to see the distribution among the three groups of patients with low CD4 nadir.
We respectfully disagree with the reviewer on this point. This in an unadjusted analysis. We believe that without adjusting for the different vaccination kinetics between the low CD4 nadir group and the other groups, this specific contrast is non interpretable and potentially misleading as it is affected by major confounding bias. For information, below we report the graph which also includes this comparison.
- The sentence in lines 87-90 “In contrast, in PLWH with low CD4 nadir the mean log2 difference between nAbsT against BA.1 and WD614G was smaller in those with current CD4 201-500 vs. those with CD4 <200 cells/mm3 [-0.80 (-1.52, -0.08); p=0.03] (Figure 1B, supplementary Table 2)” is difficult to understand. Difference between Wuhan and BA.1 neutralizing values seems not so different and follow the same trend. Please clarify this analysis and its relevance.
When simply looking at the raw data, neutralizing titres seem to follow the same trend in these groups, but after controlling for age, gender and time difference between second and third dose as well accounting for censored responses by regression adjustment a difference of 0.8 log2 was observed when comparing PLWH with a CD4 count of 200-500 vs. <200/mm3. We have slightly revised the sentence to improve clarity.
- To conclude from previous analysis that “In contrast, in PLWH with a CD4 nadir <200 cells/mm3, current CD4 count appeared to play a role in diversifying the level of SARS CoV-2 neutralization” is not justified.
We agree that this is supported by our data and modelling results only under very strong assumptions. We have added a sentence in the Discussion to acknowledge these assumptions.
- Differences in neutralization activity against original Wuhan and ómicron variants have been described in many works because the vaccine used is based on the sequence of the original SARS-CoV-2 strain and elicited antibodies recognize with less efficacy BA.1 that can be considered a escape variant. This should be discussed to explain such differences.
We have added a sentence in the Discussion section as suggested.
Minor comments
- In the abstract please revise the last sentence for better understanding.
We have rephrased the sentence as suggested.
- There is a mistake in materials and methods section (lane 12): >200 should be <200
All typing mistakes have been now amended.
Round 2
Reviewer 1 Report
I have accepted the comments. Thank you.
Accept after minor revision
Line 63: (>200/mm3, CD4 201- 500/mm3 and >500 cells/mm3)
"CD4" is not necessary.
Author Response
I have accepted the comments. Thank you.
Accept after minor revision
Line 63: (>200/mm3, CD4 201- 500/mm3 and >500 cells/mm3)
"CD4" is not necessary.
These minor modifications have been done, thank you.
Reviewer 2 Report
Regarding major comment 5, and authors´response
- In panel A of figure 1 it should be included the results of all patients included in the low nadir group for comparison. Panel B should remain to see the distribution among the three groups of patients with low CD4 nadir.
We respectfully disagree with the reviewer on this point. This in an unadjusted analysis. We believe that without adjusting for the different vaccination kinetics between the low CD4 nadir group and the other groups, this specific contrast is non interpretable and potentially misleading as it is affected by major confounding bias. For information, below we report the graph which also includes this comparison.
I have two comments regarding this point
1. I have not found the graph reported by the authors, please send it again
2. Actually vaccination kinetics is not very different from PLWH with high and low CD4 counts as stated by the authors in lanes 87-91 and supp table 1. Time interval between second and third dose is actually more close between PLWH with low and high CD4 (median 157 and 182 days respectively) than between high CD4 and HCW (182 and 283 days respectively).
Regarding point 7.
- To conclude from previous analysis that “In contrast, in PLWH with a CD4 nadir <200 cells/mm3, current CD4 count appeared to play a role in diversifying the level of SARS CoV-2 neutralization” is not justified.
We agree that this is supported by our data and modelling results only under very strong assumptions. We have added a sentence in the Discussion to acknowledge these assumptions
The sentence is:
"Interestingly, under the strong set of assumptions implicit in our model, in PLWH with a CD4 nadir <200 cells/mm3, current CD4 count appeared to play a role in diversifying the level of SARS-CoV-2 neutralization" (lanes 168-170)
I insist that is very difficult to make this point a robust conclussion from the data. If CD4 contributes to qualitative or quantitative aspects of neutralizing antibodies a difference should be found for PLWH with >500 CD4 and HCW groups with respect to the low CD4 counts groups. I suggest to keep this point on the discussion.
Author Response
Regarding major comment 5, and authors´response
- In panel A of figure 1 it should be included the results of all patients included in the low nadir group for comparison. Panel B should remain to see the distribution among the three groups of patients with low CD4 nadir.
We respectfully disagree with the reviewer on this point. This in an unadjusted analysis. We believe that without adjusting for the different vaccination kinetics between the low CD4 nadir group and the other groups, this specific contrast is non interpretable and potentially misleading as it is affected by major confounding bias. For information, below we report the graph which also includes this comparison.
I have two comments regarding this point
- I have not found the graph reported by the authors, please send it again
We had specified to the Editor, in the cover letter, to send you pdf because it was not possible to include the graph in the response platform. You will find it in the cover letter and the same graph has been included in the manuscript according with your suggestion.
Actually vaccination kinetics is not very different from PLWH with high and low CD4 counts as stated by the authors in lanes 87-91 and supp table 1. Time interval between second and third dose is actually more close between PLWH with low and high CD4 (median 157 and 182 days respectively) than between high CD4 and HCW (182 and 283 days respectively).
This is a fair point. Because participants with low CD4 count have been prioritized for vaccination they ended up to have a much longer gap between doses that expected under current guidelines and this explains the smaller difference between the low and high CD4 count group. We have now added the low CD4 cunt group in the graph as requested.
Regarding point 7.
- To conclude from previous analysis that “In contrast, in PLWH with a CD4 nadir <200 cells/mm3, current CD4 count appeared to play a role in diversifying the level of SARS CoV-2 neutralization” is not justified.
We agree that this is supported by our data and modelling results only under very strong assumptions. We have added a sentence in the Discussion to acknowledge these assumptions
The sentence is:
"Interestingly, under the strong set of assumptions implicit in our model, in PLWH with a CD4 nadir <200 cells/mm3, current CD4 count appeared to play a role in diversifying the level of SARS-CoV-2 neutralization" (lanes 168-170)
I insist that is very difficult to make this point a robust conclussion from the data. If CD4 contributes to qualitative or quantitative aspects of neutralizing antibodies a difference should be found for PLWH with >500 CD4 and HCW groups with respect to the low CD4 counts groups. I suggest to keep this point on the discussion.
We have now added the sentence in the discussion as requested and slightly modified our conclusions accordingly.